# Large bipolaron density at organic semiconductor/ electrode interfaces

Rijul Dhanker[1], Christopher L. Gray[2], Sukrit Mukhopadhyay[3], Sean Nunez[4], Chiao-Yu Cheng[1], Anatoliy N. Sokolov[3] & Noel C. Giebink [1]

Bipolaron states, in which two electrons or two holes occupy a single molecule or conjugated polymer segment, are typically considered to be negligible in organic semiconductor devices due to Coulomb repulsion between the two charges. Here we use charge modulation spectroscopy to reveal a bipolaron sheet density >$10^{10}$ cm$^{-2}$ at the interface between an indium tin oxide anode and the common small molecule organic semiconductor N,N'-Bis(3-methylphenyl)-N,N'-diphenylbenzidine. We find that the magnetocurrent response of hole-only devices correlates closely with changes in the bipolaron concentration, supporting the bipolaron model of unipolar organic magnetoresistance and suggesting that it may be more of an interface than a bulk phenomenon. These results are understood on the basis of a quantitative interface energy level alignment model, which indicates that bipolarons are generally expected to be significant near contacts in the Fermi level pinning regime and thus may be more prevalent in organic electronic devices than previously thought.

[1] Department of Electrical Engineering, The Pennsylvania State University, University Park, Pennsylvania, PA 16802, USA. [2] Department of Chemistry, The Pennsylvania State University, University Park, Pennsylvania, PA 16802, USA. [3] The Dow Chemical Company, 1776 Building, Midland, MI 48674, USA. [4] Department of Materials Science, The Pennsylvania State University, University Park, Pennsylvania, PA 16802, USA. Correspondence and requests for materials should be addressed to N.C.G. (email: ncg2@psu.edu)

Polarons are the dominant charge carrying species in disordered organic semiconductors[1]. They are typically localized on a single molecule or conjugated polymer segment and consist of an electron or hole dressed by a distortion of the intramolecular nuclear framework, as well as the electronic polarization and lattice relaxation of the surroundings[1]. Bipolarons consisting of two electrons or two holes on a single-molecular site are also possible, but are difficult to form because their Coulomb repulsion tends to outweigh the stability gained through nuclear reorganization, resulting in Hubbard formation energies $U > 0.1$ eV[1–3].

Most electrical injection and transport models consequently neglect bipolarons. However, it has been pointed out that strong energetic disorder in organic thin films may facilitate bipolaron formation in the density of states (DOS) tail (since hopping from a higher energy site to a lower energy site can compensate for $U$), and that this may in turn explain organic magnetoresistance (OMAR) in unipolar devices[2,3]. In this context, one might expect bipolaron densities to be particularly pronounced at organic semiconductor/electrode interfaces, where energetic disorder is typically enhanced relative to the bulk[4,5] and Fermi level pinning can lead to large interfacial charge densities[6–8].

Here we report the existence of a large (hole) bipolaron density located near the interface between an indium tin oxide (ITO) contact and the common small molecule hole transport material N,N′-Bis(3-methylphenyl)-N,N′-diphenylbenzidine (TPD). Using charge modulation (CM) spectroscopy[9–11] to directly monitor the density of TPD polarons and bipolarons in hole-only devices, we show that the latter can account for >1% of the interfacial charge under forward bias and that variations in the bipolaron density correlate closely with the magnetocurrent response in accord with the bipolaron model of OMAR[2]. These results are understood by generalizing the interface energetic model of Oehzelt et al.[8], which indicates that significant bipolaron concentrations are likely to be common near contacts in the Fermi level pinning regime and therefore that bipolaron-based OMAR may be more of an interfacial than a bulk effect.

## Results

**Bipolaron absorption in devices.** Figure 1a shows the cation and dication absorption spectra of TPD that has been electrochemically oxidized in dichloromethane solution; see Supplementary Fig. 1 for experimental details. Consistent with previous reports, biasing above the first oxidation potential at $V = 0.55$ V (see the inset cyclic voltammetry scan) leads to a decrease in absorbance by the neutral molecule ($\lambda = 380$ nm, black line) and to the emergence of two new bands at $\lambda = 484$ nm and $\lambda = 1400$ nm associated with the TPD cation[10,12]. Subsequently increasing the bias to 0.9 V above the second oxidation potential leads to a third absorption band centered at $\lambda = 734$ nm due to formation of the dication[12].

Figure 1b shows that the same absorption features can also be observed in a thin film of TPD co-evaporated with 10 wt% $MoO_3$. In this case, the large electron affinity of $MoO_3$ induces electron transfer from the highest occupied molecular orbital (HOMO) of TPD, p-doping the film with hole polarons and, evidently, also bipolarons[13,14]. Based on the cation and dication absorption cross-sections evaluated from their molar extinction coefficients in solution (Fig. 1a; $\varepsilon^+ = 3.8 \times 10^4$ $M^{-1}$ $cm^{-1}$ and $\varepsilon^{++} = 7.5 \times 10^4$ $M^{-1}$ $cm^{-1}$ at $\lambda = 1400$ nm and $\lambda = 743$ nm, respectively), we estimate polaron and bipolaron concentrations of $P^+ \sim 10^{20}$ $cm^{-3}$ and $P^{2+} \sim 2 \times 10^{19}$ $cm^{-3}$, respectively. While only a fraction of this polaron density is mobile (the free polaron density obtained from conductivity measurements is an order of magnitude lower)[13], this result is notable because it

shows that the bipolaron concentration can indeed be significant in organic semiconductors—roughly 20% of the polaron concentration in this case.

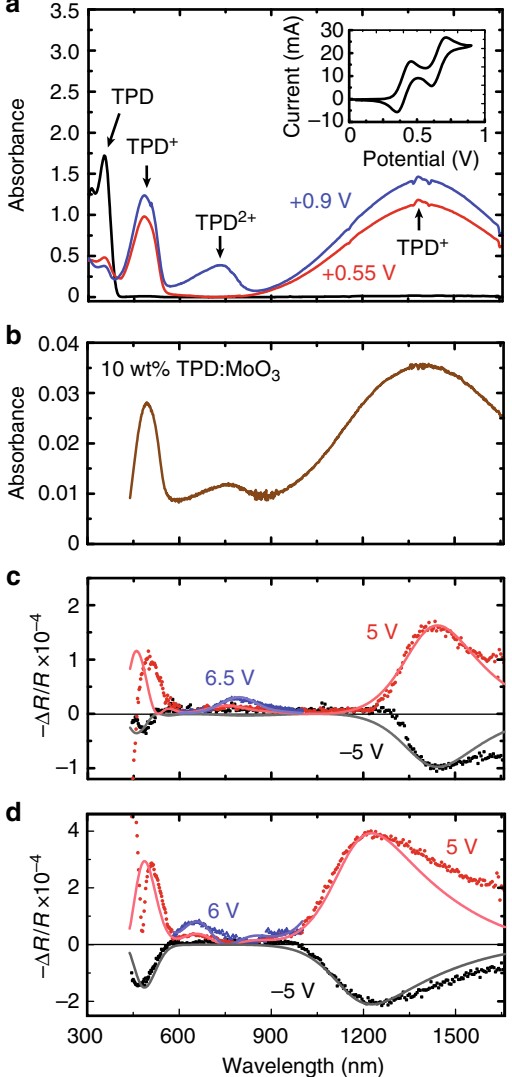

**Fig. 1** Polaron and bipolaron spectroscopy. **a** Absorption spectra of the TPD cation and dication in solution. The solution consists of 55 μM TPD dissolved in dichloromethane with a 0.5 M tetrabutylammonium tetrafluoroborate electrolyte. Electrochemically oxidizing the initially clear solution (black line) for 1 h at 0.55 V with respect to a Ag/Ag+ reference electrode causes it to turn orange due to the formation of TPD cations that absorb in the blue and near-infrared (red line). Further oxidizing the solution at 0.9 V for 2 h (blue line) causes it to turn dark green due to a new absorption band at $\lambda = 734$ nm originating from TPD dications. The cyclic voltammetry scan in the inset yields first and second oxidation potentials $E_{1/2}^+ = 0.45$ V and $E_{1/2}^{2+} = 0.71$ V, respectively. **b** Absorption coefficient of a 50 nm thick film of TPD thermally co-evaporated with 10 wt% $MoO_3$. **c** Charge modulation difference spectra recorded for an ITO (100 nm)/TPD (250 nm)/Ag (100 nm) hole-only device pulsed at −5 V reverse bias (black data points) and 5 V forward bias (red data points) using s-polarized light incident at a 45° angle. Increasing the voltage to 6.5 V (blue data points) confirms the emergence of the bipolaron absorption band at $\lambda \sim 780$ nm. **d** Similar data for a 200 nm thick device, demonstrating the optical interference-induced spectral shift that occurs for the polaron and bipolaron absorption bands. The solid lines in (**c**, **d**) are produced by a transfer matrix optical model based on the polaron and bipolaron lineshapes in (**b**) under the assumption that both species are located in a 1 nm thick layer adjacent to the ITO anode

The bipolaron absorption signature is also an evident in un-doped TPD hole-only devices (unipolar operation is verified in all devices studied here by their lack of electroluminescence as detailed in Supplementary Figure 2) as shown in the CM spectra of Fig. 1c. There, the change in reflectivity of a plasma-treated ITO (100 nm)/TPD (250 nm)/Ag (100 nm) device is monitored synchronously in response to a square-wave signal (400 Hz frequency) in both forward and reverse bias. In agreement with previous CM spectroscopy measurements on the related hole transport material 4,4′-bis[N-(1-naphthyl)-N-phenylamino] biphenyl (NPD)[10], we observe induced absorption (transparency) at $\lambda \sim 500$ nm and $\lambda \sim 1415$ nm associated with an increase in TPD polaron density under forward (reverse) bias. In the present case, however, we are also able to resolve induced absorption by the TPD bipolaron in forward bias at $\lambda \sim 780$ nm since this transition does not overlap spectrally with the polaron as in the case of NPD[14].

While the polaron and bipolaron absorption features in the device (Fig. 1c) generally coincide with those in the p-doped TPD film (Fig. 1b), their lineshapes differ (most noticeably in the case of the near-infrared polaron feature) due to interference effects originating from reflection by the Ag cathode[10]. Transfer matrix optical modeling[15] of the CM spectra indicates that both the polaron and bipolaron absorbing species must be concentrated near the ITO interface in order to fit the experimental data. This conclusion is supported by Fig. 1d, where the CM spectra for a 200 nm thick device exhibit interference-related blue shifts of the polaron and bipolaron absorption (relative to Fig. 1c) that can only be reproduced (solid lines) by assuming both to be near the ITO interface; see Supplementary Figure 3 for details. Taken together, Fig. 1 supports the original picture established by Marchetti et al.[10], where a large positive interfacial charge density exists adjacent to the ITO, but now shows that a non-negligible fraction of this charge density exists in the form of bipolarons.

**Magnetic field dependence.** Figure 2a shows the magnetic field response of the polaron and bipolaron densities in the 200 nm thick device operated at 5 V forward bias. When a strong per-manent magnet is moved close to the device (Supplementary Figure 4 provides experimental details), the bipolaron induced absorption signal at $\lambda \sim 650$ nm decreases by $\sim 40\%$ and the polaron induced absorption signal at $\lambda \sim 1230$ nm increases by $\sim 15\%$. These changes coincide with a $\sim 7\%$ magnetoresistive decrease in the current superimposed upon a slow background drift that occurs naturally over the course of the experiment. Although we do not know the precise magnetic flux density within the device (estimated to be >0.5 T), the sign of each change is consistent with bipolaron-based OMAR, where the magnetic field causes a spin blockade for bipolaron formation (thus shifting the balance toward more polarons) that reduces the number of percolation paths for charge transport[2,3,16–18].

Figure 2b shows that the magnetoresistance increases with applied bias in direct proportion to the bipolaron concentration (relative to zero bias, i.e., $\Delta P^{2+} = P^{2+}(V) - P^{2+}(0)$), which is calculated from the induced absorption at $\lambda \sim 650$ nm using the transfer matrix optical model and assuming that the bipolarons exist in a 1 nm thick sheet adjacent to the ITO. The polaron density ($\Delta P^+$, similarly obtained from the induced absorption at $\lambda \sim 1230$ nm) displays a weaker voltage dependence in Fig. 3a, but does correlate with the overall magnitude of the magnetoresistive response as the effective work function of the ITO anode is varied by adding a 4 nm layer of $MoO_3$ or neglecting the plasma treatment in otherwise identical devices. Although the bipolaron CM signal in these latter cases is below our detection limit, it is reasonable to expect that the bipolaron density varies

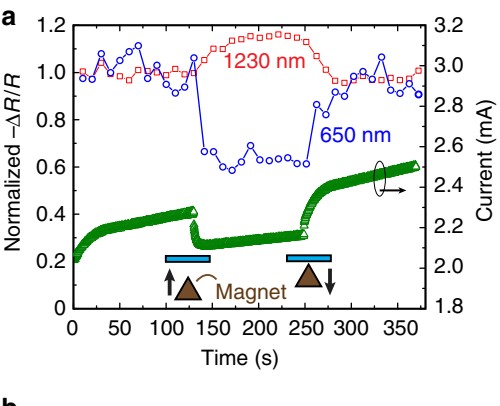

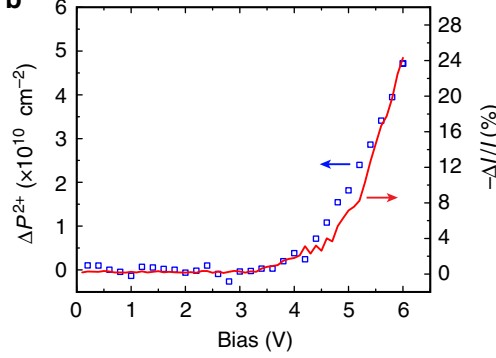

**Fig. 2** Magnetic field response. **a** Magnetic field response of the polaron and bipolaron CM absorption bands recorded at $\lambda = 1230$ nm and $\lambda = 650$ nm, respectively, from a 200 nm thick TPD device operated at 5 V forward bias. The magnetic field is applied during the indicated time interval by moving the apex of a flux-concentrated pyramidal permanent magnet adjacent to the device cathode as illustrated in the inset. The right-hand axis shows the associated decrease in current that occurs upon application of the magnetic field, which is estimated to be >0.5 T. **b** Bias dependence of the interfacial bipolaron concentration (blue squares) calculated from optical modeling of the charge modulation spectra. The right-hand axis shows that the magnetocurrent response of the device grows in direct proportion to the bipolaron concentration

monotonically with the polaron density given that the former is a bimolecular product of the latter[19]. To this point, the bipolaron-to-polaron ratio ($P^{2+}/P^+$) in the plasma-treated ITO device increases with bias, with the bipolaron concentration reaching $\sim 1\%$ of the total polaron concentration at 6 V as shown in the inset; this represents a lower bound based on the maximum total polaron concentration estimated in Supplementary Figure 5.

The linear variation in polaron density observed for $V < 3$ V is roughly a factor of five less than that calculated from the geometric capacitance, which suggests that only a fraction of the capacitive electrode charge transfers in and out of the first TPD monolayer(s)[10]. Notably, this fraction depends on the ITO surface treatment. The fact that plasma treated ITO yields a larger swing in the number of charged TPD molecules than untreated ITO for the same geometric capacitance and voltage swing indicates that the density of interface states at the Fermi energy ($E_F$) must be higher in the former case. This follows straightfor-wardly from the higher work function of plasma-treated ITO[20], which shifts $E_F$ down, closer to the peak of the disorder-broadened HOMO interface DOS[8].

This work function shift unsurprisingly leads to a dramatic difference in current density between the two devices as shown in Fig. 3b, where the diffusion current stemming from the large interfacial charge density in the plasma-treated device is clearly evident from the exponential increase at $\sim 1$ V bias[21]. The lower

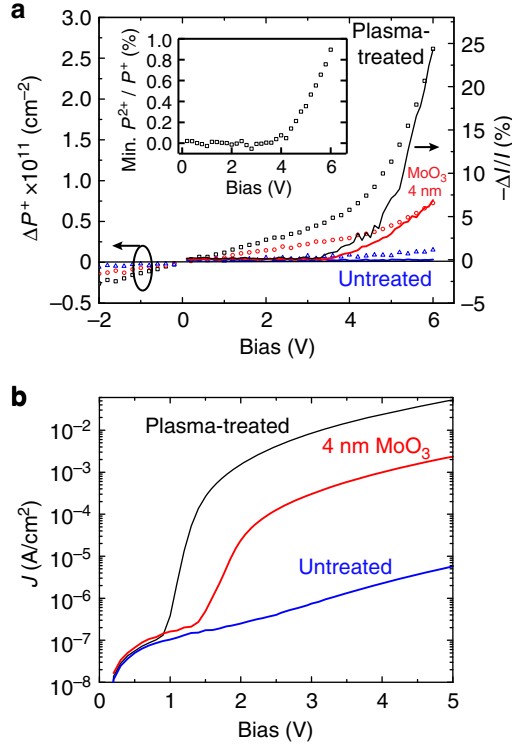

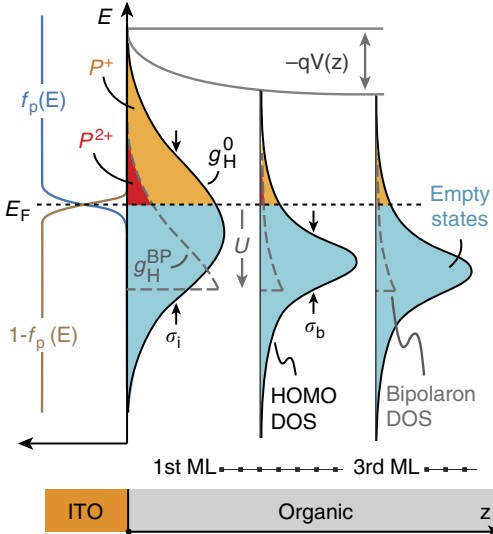

**Fig. 4** Bipolaron interface energetic model. Diagram of the interfacial energy level alignment model. A Gaussian HOMO density of states ($g_H^0$) is assumed for each monolayer of the organic, with different broadening assumed for the interface ($\sigma_i$ in the first monolayer) and the bulk ($\sigma_b$ in subsequent monolayers). States above the Fermi level that are occupied by holes ($P^+$, orange shading) subsequently become possible sites for bipolaron formation at lower energy, leading to the gray-dashed bipolaron DOS shifted down by $U$. The states in $g_H^{BP}$ that lie above $E_F$ become bipolarons ($P^{2+}$, red shading)

**Fig. 3** Impact of anode surface treatment. **a** Bias-dependent change in interfacial polaron concentration, $\Delta P^+$, calculated for a series of devices with different ITO surface treatments based on their associated charge modulation spectra. The polaron concentration is maximum for plasma-treated ITO (black squares) and becomes much smaller without the plasma treatment (blue triangles). Adding a 4 nm thick $MoO_3$ layer leads to an intermediate polaron concentration (red circles). The right-hand axis shows that the magnetocurrent response of each device trends with the overall change in polaron concentration between them; however, the magnetocurrent exhibits a stronger voltage dependence than the polaron density within any given device. The inset shows the minimum absolute bipolaron-to-polaron ratio ($P^{2+}/P^+$) estimated for the plasma-treated ITO device as a function of bias; the bipolaron CM signal for the other devices was below the detection limit. **b** Current–voltage relationship measured for each device

current density of the $MoO_3$-based device is somewhat surprising, though not unexpected given the 4 nm thickness employed here[22], and is nonetheless consistent with the intermediate interfacial charge density determined in Fig. 3a. Taken together, these data point to a proportional relationship between magnetoresistance and interfacial bipolaron density (Fig. 2)[2], and suggest that both quantities increase with the interfacial polaron density as the effective anode work function increases (Fig. 3).

**Interfacial bipolaron model**. To understand the nature of the interfacial charge density in more detail, we generalize the electrostatic energy level alignment model of Oehzelt et al.[8] to include bipolaron occupation statistics. Figure 4 outlines the model, where each monolayer of the organic semiconductor is characterized by a Gaussian HOMO DOS distribution, $g_H^0(E)$, defined by its peak energy ($E_H$) and standard deviation, which may be different at the interface ($\sigma_i$ for the first monolayer) than in the bulk ($\sigma_b$ for all subsequent monolayers)[4,5]. As the Fermi level of the system equilibrates, charge is transferred between the electrode and organic layers, giving rise to a spatially varying charge

density [$\rho(z)$] and consequent potential shift [$V(z)$] in accord with the Poisson equation.

Acknowledging the possibility of bipolaron formation subject to a Hubbard energy penalty, $U$, means that singly occupied HOMO states in $g_H^0(E)$ generate an additional density of states for bipolaron occupation given by:

$$g_H^{BP}(E) = f_p(E + U)g_H^0(E + U), \qquad (1)$$

where $f_p(E) = 1 - [1 + \exp(E - E_F)/k_bT]^{-1}$ is the usual Fermi-Dirac occupation function for holes. This description is related to, but different than that given in Ref. [3].

The charge density in each layer subsequently follows from integrating over the singly occupied states, as well as those that gain an additional hole to become bipolarons:

$$\rho(z) = qN_{mol} \int f_p(E)\left[g_H^0(E) + \chi_s g_H^{BP}(E)\right]dE, \qquad (2)$$

subject to the probability, $\chi_s(B)$, that they adopt a singlet spin configuration. We assume $\chi_s = 1$ for the sake of simplicity since, in the absence of an external magnetic field (i.e., $B = 0$), the spin of each polaron interacts with randomly oriented hyperfine fields, enabling triplet pair states to eventually transform into singlets within the typical dwell time that two polarons spend next to one another[2,3]. The bipolaron density is thus $P^{2+} = N_{mol} \int f_p(E)\chi_s g_H^{BP}(E)dE$ and the polaron density is $P^+ = N_{mol} \int f_p(E)\left[g_H^0(E) - \chi_s g_H^{BP}(E)\right]dE$. In principle, electron polaron ($N^-$) and bipolaron ($N^{2-}$) charge density in the lowest unoccupied molecular orbital (LUMO) should also be included in $\rho(z)$[8], though we neglect it in Eqn. 2 for clarity since the electron density is negligible in the case at hand (i.e., when $E_F$ is close to the HOMO).

This model is solved iteratively as described in Ref. [8] for the ITO/TPD system using representative parameters obtained from ultraviolet photoelectron spectroscopy (ITO work function

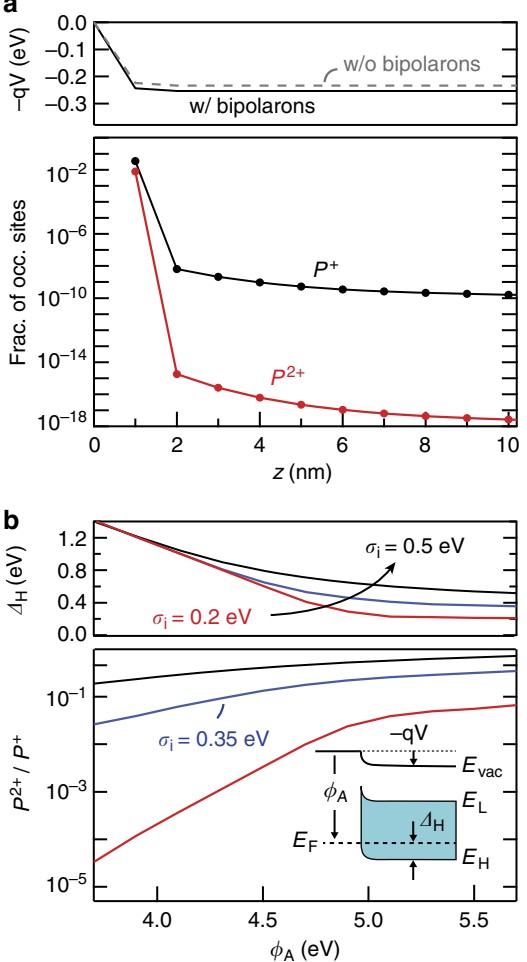

**Fig. 5** Bipolaron model predictions. **a** Polaron and bipolaron densities (lower panel) and potential shift (upper panel) calculated for each monolayer of TPD assuming $\sigma_i = 0.35$ eV, $\sigma_b = 0.1$ eV, an ITO work function $\phi_A = 4.9$ eV, and a TPD ionization potential $IP = 5.1$ eV (defined as the onset of $g_H^0$, two standard deviations less than its peak at $IP + 2\sigma_b = 5.3$ eV). The density of TPD molecules is assumed to be $10^{21}$ cm$^{-3}$, with each monolayer being 1 nm thick. The gray dashed line in the upper panel shows the potential shift calculated for the same system without the possibility of bipolaron formation. **b** The lower panel shows the bipolaron-to-polaron ratio in the first monolayer calculated for varying anode work function and different $\sigma_i$. The upper panel shows the associated variation in hole injection barrier shown in the inset ($\Delta_H$ is defined as the difference between $E_F$ and the HOMO onset in the bulk), where Fermi level pinning occurs at $\phi_A > 5$ eV

$\phi_A = 4.9$ eV, TPD ionization potential $IP = 5.1$ eV)[23] and a Hubbard energy $U = 0.25$ eV estimated from the difference between the first and second TPD oxidation potentials shown in the inset of Fig. 1a[12]. We take $\sigma_b = 0.1$ eV based on bulk transport measurements of the closely related molecule NPD[24,25] and $\sigma_i = 0.35$ eV in accord with estimates of the interfacial distribution obtained from internal photoemission and impedance spectroscopy[26,27].

Figure 5a displays the calculated potential shift (top panel) and carrier density profiles (lower panel) given as a fraction of the occupied site density in each monolayer ($N_{mol} = 10^{14}$ cm$^{-2}$). The vast majority of the charge resides in the first monolayer, giving rise to a ~0.3 eV interfacial dipole as discussed previously[8] and observed experimentally[23]. Importantly, bipolarons constitute a significant fraction of this interface charge ($P^{2+}/P^+ = 0.22$) but are

negligible in the bulk ($P^{2+}/P^+ < 10^{-7}$). The existence of this substantial bipolaron charge density does not, however, lead to an appreciable change in the energy level alignment as shown by the dashed potential distribution in the upper panel, which is calculated without the possibility of bipolaron formation (i.e., in the limit $U \to \infty$). In this case, the loss of bipolaron charge density is largely compensated by an increase in the polaron density as dictated by the electrostatics of the system.

Apart from the Hubbard energy, the primary factors that influence the bipolaron density in this model are the width of the interface DOS ($\sigma_i$) and the position of the Fermi level relative to the HOMO (dictated by the ITO work function, $\phi_A$). The lower panel of Fig. 5b shows that the bipolaron-to-polaron ratio ($P^{2+}/P^+$) in the first monolayer increases strongly with $\sigma_i$ and saturates with $\phi_A$ in the Fermi level pinning regime. Fermi level pinning above $\phi_A \sim 5$ eV is evident from the associated hole injection barriers ($\Delta_H$, depicted in the inset) plotted in the top panel, which transition from a slope $S = -d\Delta_H/d\phi_A = 1$ (Mott–Schottky limit) toward $S \to 0$ (Fermi level pinning)[8]. The main conclusion from Fig. 5b is that, in the Fermi level pinning regime, bipolarons can generally be expected to make up a substantial fraction of the interfacial charge density ($P^{2+}/P^+ > 0.01$) over the entire range of $\sigma_i$ that have been estimated to date[4,26–28]. This fraction is even higher for alternative interface DOS distributions, such as Lorentzian or exponential functions that decay more slowly than the Gaussian employed here[8,28].

## Discussion

The nature of the TPD bipolaron state itself also warrants discussion. Density functional theory calculations (B3LYP/6-31 g** using a background dielectric constant $\varepsilon_r = 2.5$; see Supplementary Figure 6) show that two different electronic states are possible for the bipolaron: a closed-shell singlet dication state where both holes delocalize over the whole molecule, and a diradical singlet state where the holes are localized at opposite ends[29–32]. Computations for a single isolated TPD molecule suggest that the Hubbard energy is lower for the diradical than the closed-shell singlet state by roughly 0.1 eV, and that this difference becomes more pronounced with increasing dihedral twist angle of the bridging biphenyl moiety. The diradical state may therefore be the dominant species observed in our experiments; however, we cannot make a spectroscopic distinction at this time since both are predicted to exhibit similar absorption spectra based on time-dependent density functional theory (see Supplementary Figure 7).

In conclusion, we have shown that a large TPD bipolaron density exists near the interface with an ITO anode and have provided the first direct evidence associating unipolar magnetoresistance with variation in the bipolaron concentration. Electrostatic modeling of the interface energetics suggests that bipolarons are generally expected to be significant (i.e., >1% of the polaron concentration) for energetically broad interface DOS distributions at contacts in the Fermi level pinning regime. In this context, bipolarons may also be significant at other interfaces that sustain high charge density in a broadened interfacial DOS, such as the channel of organic thin film transistors[19,33], which might be exploited to create devices with new magnetic functionality.

## Methods

**Computational modeling.** Density functional theory (DFT) calculations were carried-out using the B3LYP functional, where Becke's three-parameter hybrid exchange functional is combined with the Lee–Yang–Parr correlation functional[34], with a 6–31 G** split valence plus polarization basis set. The effect of the surrounding medium is incorporated by using a conductor-like polarizable continuum model (cpcm), where the dielectric constant of the medium is chosen to be 2.5. The excited states were computed using time-dependent DFT at the same level, with

fixed nuclear geometries maintained from the respective ground states. All DFT calculations were performed with the G09 suite of programs[35].

**Fabrication.** Devices were prepared on pre-patterned ITO-coated glass (100 nm, 20 Ω/□) substrates that were solvent-cleaned and treated with air plasma for 5 min before loading into a thermal evaporator with a base pressure of ~$10^{-7}$ Torr. Gradient sublimation-purified TPD was deposited at a rate of 0.3 nm s$^{-1}$ followed by a 100 nm thick Ag cathode contact defined via a shadow mask to yield 2 × 2 mm$^2$ individual device areas. Devices were encapsulated in a nitrogen-filled glove box with a thin layer of Norland optical adhesive.

**Characterization.** Cyclic voltammetry measurements were performed using a Metrohm Autolab PGSTAT 128 N potentiostat/galvanostat on a solution of 55 µM TPD with 0.5 M tetrabutylammonium tetrafluoroborate dissolved in dichloromethane. Platinum wire mesh working and counter electrodes were used with a Ag/Ag$^+$ standard reference electrode in an H-cell with chambers separated by a glass frit with 10–20 µm size pores. The Ag/Ag$^+$ reference electrode consisted of 0.01 M AgNO$_3$ and 0.1 M tetrabutylammonium tetrafluoroborate electrolyte in acetonitrile. Absorption spectra were measured relative to an electrolyte-solvent reference by pipetting solution from the electrochemical cell into quartz cuvettes for analysis in a Varian Cary 6000i spectrophotometer. Supplementary Figure 1 provides additional experimental details.

Charge modulation spectroscopy measurements were performed using an Energetiq laser-driven Xe light source filtered through a monochromator with order sorting filters. The light was s-polarized using a wire grid polarizer and was incident on devices through the glass substrate at an angle of 45°. The relative change in reflectivity (ΔR/R) due to a 400 Hz square wave voltage signal was detected synchronously using Si and Ge photodiodes and a lock-in amplifier. Magnetic field-dependent measurements were conducted by repeatedly moving a permanent neodymium pyramid magnet adjacent to, and then away from, a given device and monitoring the resultant change in current and/or reflectivity as detailed in Supplementary Figure 4.

**Data availability.** The data that support the findings of this study are available from the corresponding author upon reasonable request.

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

## Acknowledgements

This work was supported in part by the Dow Chemical Company as part of the Flexible Electronics UPI program and by the U.S. Department of Energy, Office of Basic Energy Sciences under Award DE-SC0012365.

## Author contributions

R.D. fabricated the samples, carried out the measurements, and performed the data analysis. C.L.G. assisted with the spectroelectrochemistry measurements and S.M.

performed the density functional theory modeling. A.N.S. and N.C.G. supervised the work. All authors contributed in writing the manuscript.

## Additional information

**Competing interests:** A.N.S. and S.M. are involved with commercial electronic materials research at the Dow Chemical Co.

