## [Peer Review File · Nature Communications]

REVIEWERS' COMMENTS:

Reviewer #1 (Remarks to the Author):

A nice paper highlighting the importance of charge transfer at interfaces and in particular the formation of bipolaronic charged species. The paper is well written and easy to follow. I have only one minor issue, the authors attribute disorder at the interface that broadens the HOMO (and LUMO) features as the facilitator/reason for charge transfer mainly to the first layer at the interface. Indeed, the intermolecular order is likely different at the interface than in the bulk, but this disorder is not expected to end at the first layer but will be "passed on" to several subsequent layers until the bulk intermolecular order is achieved and hence extended "band bending" should follow, which typically is not the case. A second and often more important reason for charge transfer being primarily located in the first layer is the electrostatic interaction with the substrate which shifts the oxidation and reduction energies compared to the bulk, see e.g. *Advanced Functional Materials* 26 (2016) 1077 and *Advanced Materials* 29 (2017) 1606901.

Reviewer #2 (Remarks to the Author):

Bipolarons have been a part of pi-conjugated polymer physics since nearly the very beginning of that field. However, these states are often considered to be of negligible importance in organic semiconductor devices due to Coulomb repulsion between the two charges. More recently, bipolarons have been postulated to be important to explaining the positive magnetoresistance in unipolar organic devices, but experimental proof has remained elusive. The present manuscript provides strong evidence for this. Various spectroscopies are used to show that a significant bipolaron density exists near the electrode. Moreover, the magnetocurrent response of hole-only devices correlates closely with changes in the bipolaron concentration, supporting the bipolaron model of unipolar organic magnetoresistance. A thorough theoretical treatment is also provided. These are important advances, and I support the publication of this manuscript. The only problem I have with the manuscript is the way that magnetoresistance is measured in the devices, by moving a permanent magnet near the device rather than sweeping the magnetic field using an electromagnet. Compared to the otherwise high level of sophistication of the manuscript, this method seems very unsophisticated and below common standards for magnetoresistance measurements. But because the remaining work appears solid, I am willing to overlook that.

Review Response for Manuscript Entitled:

Large bipolaron density at organic semiconductor-electrode interfaces

R. Dhanker, C.L. Gray, S. Mukhopadhyay, S. Nunez, C-Y. Cheng, A. Sokolov, and N.C. Giebink

We appreciate the informative feedback and constructive criticism from both of the reviewers. We have addressed each comment/question as detailed below. Original reviewer comments are listed in *italics*, our responses are in black, and revisions in the manuscript are highlighted in yellow.

Response to reviewer # 1:

I have only one minor issue, the authors attribute disorder at the interface that broadens the HOMO (and LUMO) features as the facilitator/reason for charge transfer mainly to the first layer at the interface. Indeed, the intermolecular order is likely different at the interface than in the bulk, but this disorder is not expected to end at the first layer but will be "passed on" to several subsequent layers.

We fully agree that the disorder is likely to evolve over the first few monolayers toward that in the bulk. Our model is fully compatible with this reality, as a different density of states can be chosen for each monolayer. However, to minimize the number of parameters in the model as well as speculation about the unknown functional dependence of the transition toward the bulk density of states (i.e. does it occur in the first two monolayers vs. the first five, etc.), we have adopted the approach of Oehzelt et. al. (Ref. 8 in the manuscript) and simplified the situation to just two individual disorder widths for the interface and bulk.

Responses to reviewer #2:

The only problem I have with the manuscript is the way that the magnetoresistance is measured in the devices, by moving a permanent magnet near the device rather than sweeping the magnetic field using an electromagnet. Compared to the otherwise high level of sophistication of the manuscript, this method seems very unsophisticated and below common standards for magnetoresistance measurements.

Our use of a permanent magnet is due to the fact that we do not have an electromagnet appropriate for making organic magnetoresistance measurements in our laboratory. While we would of course prefer to use an electromagnet for greater control and variation of the field strength, the use of permanent magnets is not uncommon for organic magnetoresistance measurements in the literature (see, e.g. Song *et al. Phys. Rev. B* **82**, 085205 (2010)) and is justified in the present case since our focus is limited to a qualitative field on vs. field off type of response.